# I'm fine with collecting data: Engagement profiles differ depending on scientific activities in an online community of a citizen science project

Till Bruckermann[1,2]*, Hannah Greving[3], Milena Stillfried[4], Anke Schumann[4], Miriam Brandt[4], Ute Harms[2]

1 Institute of Education, Leibniz University Hannover, Hannover, Germany, 2 IPN–Leibniz Institute for Science and Mathematics Education, Kiel, Germany, 3 Leibniz-Institut für Wissensmedien (IWM), Tübingen, Germany, 4 Leibniz Institute for Zoo and Wildlife Research (IZW), Berlin, Germany

* till.bruckermann@iew.uni-hannover.de

**Data Availability Statement:** All relevant data are within the paper and its Supporting Information files.

## Abstract

Digital technologies facilitate collaboration between citizens and scientists in citizen science (CS) projects. Besides the facilitation of data transmission and access, digital technologies promote novel formats for education in CS by including citizens in the process of collecting, analyzing, and discussing data. It is usually assumed that citizens profit more from CS the more they participate in the different steps of the scientific process. However, it has so far not been analyzed whether citizens actually engage in these steps. Therefore, we investigated citizens' actual engagement in different scientific steps online (i.e., data collection and data analysis) in two field studies of a CS project. We then compared them with other CS projects. We analyzed behavioral engagement patterns of $N = 273$ participants with activity logs and cluster analyses. Opportunities to engage in different steps of the scientific process increased participants' overall commitment compared to contributory CS projects. Yet, despite their increased commitment, participants' engagement was only more active for data collection but not for data analysis. We discuss how participants' perceived role as data collectors influenced their actual engagement in the scientific steps. To conclude, citizens may need support to change their role from data collectors to data inquirers.

## Introduction

Citizen science (CS), the scientific collaboration between volunteer citizens and full-time scientists, is becoming increasingly popular. A vast number of CS projects allow citizens to participate directly in scientific activity (e.g., Zooniverse projects: [1]). Simultaneously, CS has become an invaluable research tool in many disciplines [2]. Without citizens' help in data collection, many research projects would not be possible. Citizens also benefit from engaging in CS projects. By participating in the whole scientific process, citizens have the opportunity to increase their knowledge about science and the processes of science [3,4]. Even though the

**Funding:** This work was supported by the German Federal Ministry of Education and Research (BMBF; https://www.bmbf.de) under grant numbers 01|O1725 (MB), 01|O1727 (UH), 01|O1728 (Joachim Kimmerle). The publication of this article was funded by the Open Access Fund of the Leibniz Universität Hannover. The funders had no role in study design, data collection and analysis, decision to publish, or preparation of the manuscript.

**Competing interests:** The authors have declared that no competing interests exist.

benefits of CS for citizens seem to be clear at first sight, little is known about whether citizens are actually interested in learning about the scientific process or actually *use* the opportunity to learn more about it. While there has been some research on citizens' engagement in CS projects [2,5–8], no analysis has been done so far *on the extent* to which citizens engage with the different tasks of the scientific process [9]. Therefore, we investigated to what extent citizens actually engaged in two different steps of the scientific process (i.e., data collection and data analysis) in a CS project that combined field activities with activities in an online community. Conducting this investigation provided some highly relevant insights about citizens' actual engagement patterns against the background of the ideal picture of CS projects.

## Citizen science and the digital age

Citizen science has a long tradition and, in its current form, was established at the beginning of the 20th century with the Christmas Bird Count project [10]. On the one hand, CS projects provide *scientists* with the opportunity to collect larger datasets in time and space than would be possible without volunteer contributions. Citizens' contributions to CS projects also have a high financial value in the scientific research of many disciplines [2]. Without this kind of voluntary participation, many research projects would not be financially feasible. On the other hand, CS projects give *citizens* the possibility to learn about new topics and science. This means that citizens can gain topical knowledge they may otherwise not have access to by engaging in such projects. Furthermore, they can acquire scientific knowledge, as engaging in CS projects allows them to gain insights into the whole scientific process (i.e., formulating research questions, designing a study, collecting relevant data, analyzing the data, and interpreting and discussing the results). In summary, both scientists and citizens benefit from their joint engagement in CS projects.

Nowadays, digital devices facilitate collaboration between citizens and scientists. Data can be more easily collected and transmitted via mobile devices such as smartphones or tablets. At the same time, databases are constantly available online for both scientists and citizens who can easily interact with them. Besides the facilitation of data collection, transmission, and access [11], the digital age also brings new aspects to CS projects. First, digital technologies and software enable the formation of online CS communities [11] which collaboratively develop and create knowledge through interactions of citizens or between scientists and citizens [12,13]. Second, computer-supported collaborations of citizens and scientists promote novel formats for education in CS [14] by including citizens in the process of gaining evidence from data and discussing their results online. For example, new statistical tools can enable citizens themselves to analyze data and discuss the results online [15]. At the same time, technologies also have to be designed in such a way as to facilitate data inspection, analysis, and discussion [16]. Such a scaffolding design provides assistance during the employment of tools for inquiry and helps non-experts to accomplish tasks that otherwise would be too difficult [16,17]. One can also enter a workspace that helps citizens to structure relevant data and to organize questions and results (artifacts) from data analysis [17]. Thus, digital devices, software, and tools can all facilitate citizens' engagement in CS and enable them to participate in the process of scientific research.

## Citizens' engagement in the scientific process and its challenges

Engaging citizens in the whole scientific process of CS projects has been shown to be beneficial for citizens' knowledge about science [18,19]. In order to specify which type of engagement or participation would be most beneficial for a person who wants to volunteer, several frameworks have been proposed which describe different levels of participation [19,20]. The

frameworks classify participation according to the level of citizens' inclusion in the different steps of the scientific process: The mere collection of data in a CS project is regarded as the lowest level of participation (contributory projects: [19]); collecting data, data analysis, and the interpretation of results in a CS project is seen as an intermediate level (collaborative projects: [19]), because some projects allow for data analyses that go beyond species identification activities [21] and may include statistical data analyses (e.g., [22]); and being involved in the whole research process, from generating research questions to drawing conclusions, is regarded as the highest level (co-created projects: [19]; see also 'extreme CS': [20]). Co-created projects are known to increase the cognitive engagement of citizens [20]. Other participation models also suggest that higher degrees of participation (i.e., developing explanations and discussing results) lead to higher engagement in the project [4,19]. In science education, the *model-of-data theory* on reasoning suggests that authentic inquiry tasks stimulate reasoning processes on higher cognitive levels [23,24]. Hence, including citizens in scientific activities that require reasoning and drawing conclusions from data promotes cognitive processes and, thus, increases their scientific knowledge.

However, even though citizens should be objectively able to participate in the whole scientific process in CS projects [25], it is unclear whether citizens equally engage in all levels of scientific activities or whether they are even interested in doing so [4,9,20]. First, CS projects often take place in informal learning settings [26] which provide citizens with different opportunities to learn from different scientific activities [27]. In this case, their choice of activities may not correspond to the ideal engagement pattern of the highest level [9,19]. Second, the spreading of CS in various disciplines [28] has led to CS projects offering discipline-specific activities for citizens to participate in [21]. For example, in the biological sciences, CS projects often engage participants in observation and monitoring activities [21], while in the health sciences, CS projects may involve their participants in more gamified activities such as digitally folding, and modifying proteins and their structures [29]. Such discipline-specific activities of CS projects might promote that participants' engagement differs between projects from different disciplines [1].

Third, the engagement is often described only in terms of data contribution [2,6], but does not take into account any engagement in further activities of the scientific process [9]. Recent research has extended these analyses to facilitating participants' own investigations that included analyzing data, they collected themselves, on an online platform [5]. Results showed that participants stayed connected to the project for a longer time than participants in other CS projects (i.e., Galaxy Zoo and Milky Way: [6]) in which they mostly worked on scientists' data. Third, some theoretical work [20] and empirical findings [9] also question whether citizens indeed participate equally in different phases of the scientific process, for example, in both the data collection and the data analysis phase. Thus, it remains unclear whether the mere provision of opportunities to engage in different steps of the scientific process is sufficient to actually *engage* citizens in these steps. Therefore, we investigated actual behavioral patterns of citizens in the online community of a CS project across these two different steps of the scientific process. With this investigation, we aimed to increase the understanding of citizens' actual scientific behavior, which would have important implications for designing new CS projects.

## The current research

The objective of the research presented here was to analyze the engagement patterns and profiles of participants in the online community of a CS project. Before doing this, we analyzed in a first step the engagement patterns across the whole project and compared them with engagement patterns of other projects. This comparison allowed us to relate the CS project we

investigated to three other CS projects. In the next step, we then analyzed participants' specific engagement patterns within our CS project (i.e., comparing data collection and data analysis) by means of log file analyses and cluster analyses. We focused on these forms of analyses, because previous analyses may have been confounded: Most analyses so far have focused on contributory or participatory CS projects without analyzing participants' engagement in the data collection and data analysis phases separately [5,6]. Obviously, the design for participation may change behavioral patterns and may manifest itself in different engagement profiles. We suggest that tracking citizens' actual behavioral patterns through log file analyses is beneficial, as these analyses allow for the adoption of a design which fits citizens' needs and abilities [30].

Our analyses are based on previous research [5,6] and provide a comprehensive analysis of the same engagement metrics used in these previous studies. We expand on these studies in two ways: We provide (1) a between-projects comparison of engagement metrics for CS online communities focused on data contribution or higher levels of participation, and (2) a within-project comparison of citizens' engagement with scientific activities either focused on data collection or data analysis. Based on our considerations that the facilitation of participants' own investigations enhances their commitment and that participants' engagement with higher-level scientific tasks (i.e., data analysis) is lower, we stated the following hypotheses: Participants engage for longer periods and more regularly in our collaborative CS project than participants in other contributory projects (Hypothesis 1). Participants in our CS project engage more actively in the data collection phase than the data analysis phase (Hypothesis 2).

## Method and materials

### Participants

Citizens from a metropolitan city in the East of Germany applied for participation in two different field studies of an urban ecology CS project (for more details see Project and procedure). Citizens were selected for participation based on where they lived in the city. This method was chosen because the project design strived for an equal distribution of participants across the whole city in both field studies. Those citizens who were selected as participants had access to the CS project via an online platform. On this platform, all of the participants' activities were tracked in log files. Participants could, however, withdraw from tracking by either choosing to opt-out or to use anonymous browsing. Across both field studies, data of 382 participants was available, among which 77 participants withdraw from data tracking. 32 participants with fewer than two active days were excluded for methodological reasons [6]. Thus, across both field studies, the log file data of $N = 273$ participants were ultimately analyzed ($N_1 = 141$, 74 female, 64 male, $M_{age} = 52.88$, $SD = 12.91$; $N_2 = 132$, 74 female, 56 male, 1 diverse, $M_{age} = 53.85$, $SD = 11.56$; four participants did not provide demographic information). Participants had given their written informed consent of participation in a research study and an ethics committee approved the questionnaire (LEK 2018/062).

### Project and procedure

The study reported here was part of an interdisciplinary research project on knowledge transfer in CS dealing with urban ecology in a metropolitan city in the East of Germany. Our study focused on two field studies of one CS project within this larger research project. Both field studies of the CS project were concerned with terrestrial wildlife in one of the cities and were called 'Wildlife Researchers'. For the sake of clarity, we refer to them as 'Wildlife Researchers 1' and 'Wildlife Researchers 2'. They were conducted in fall, 2018, and spring, 2019, respectively. In the field studies, participants set up camera traps in their gardens for one month to capture photographs of wildlife moving in front of the camera during the day and at night.

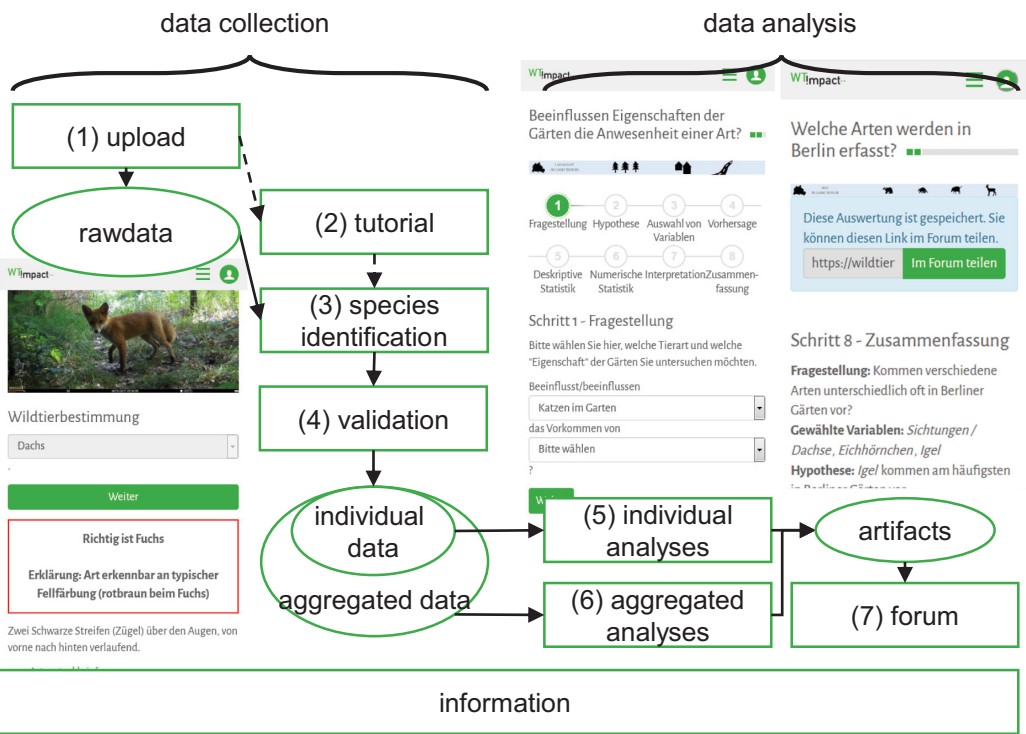

**Fig 1. Flow chart of scientific activities and resources (rectangular boxes) as well as data and artifacts (oval boxes) in the project.**

Apart from the installment and maintenance of the camera traps, all project activities took place on an online platform. We used this platform to track participants' activities in the CS project. On the online platform, participants formed an online community to share data and discuss analyses with the other participants of the respective field study of the CS project. In this community, they shared the wildlife photographs from their camera trap in a database. The photographs were used to investigate the distribution of wildlife species across the city as a function of different urbanization factors (i.e., biotic and abiotic factors). On the platform, participants could also analyze data and discuss their results and other topics in the forum. In addition, they had access to technical and methodological information about camera traps in ecological research, as well as scientific information about terrestrial mammals and urban ecology.

Fig 1 illustrates the structure of the online platform and participants' tasks within the online community: During the *data collection* phase, participants (1) uploaded camera trap photographs, (2) learned how to identify the wildlife species in the photographs from a tutorial, and (3) identified the species. Participants also (4) identified species in pictures from other participants in order to validate other participants' identification of species. These tasks of the data collection phase correspond to a task typology that clearly distinguishes observations, species identification, and data entries from data analysis [21] (e.g., statistical data analysis: [22]), although species identification could be seen as some kind of initial data analysis [31].

During the *data analysis* phase, participants (5) analyzed their individual data on the species' occurrence in their own garden and (6) aggregated data of all participants in the respective field study. Participants were guided through the analyses in a structured way using the scientific process, from the research question and hypothesis to the selection of variables and

predictions, on to descriptive and numeric statistics, and finally, to interpretation. From their analyses, participants obtained artifacts such as graphical representations and verbal descriptions of their hypotheses, predictions, and explanations of data. These artifacts were (7) made available to be shared in posts or comments in the forum for discussions with other participants and scientists. Thus, participants' tasks within the data collection were characteristic for contributory CS (simple collection and identification of photographs), while their tasks during the data analysis were characteristic for collaborative projects (analysis and interpretation of data: [19]).

## Data analysis and measures

Participants' activities (such as picture contributions, species identification, species validation, data analyses, forum posts) and their logins to the platform were saved in log files using Matomo (v3.9.1). These log files facilitated the analysis of behavioral engagement patterns and different user profiles (e.g., in the context of CS: [5]; for a review see [32,33]). Therefore, in a first step, we calculated engagement metrics in both field studies separately, based on suggestions from previous research. We then compared them to three previous CS projects, for which data on the same engagement metrics were available: Weather-it [5], Milky Way, and Galaxy Zoo [6]. Milky Way and Galaxy Zoo are contributory CS projects, while Weather-it is a collaborative project and facilitates the collection and analysis of participants' own data like our CS project did. We also used these metrics to compare participants' engagement in the data collection and data analysis phases in both field studies. In a second step, we identified user profiles based on the engagement metrics in the first field study of our CS project through cluster analysis, and we compared them for the data collection and data analysis phases separately. An analysis of transitions showed the flow of participants between the data collection and data analysis phases. The user profiles of the second field study were not calculated, due to a lower number of participants that did not suffice for further cluster analyses.

**Engagement metrics.**  We based the calculation of engagement metrics from log files data on previous suggestions for online communities in CS [5,6]. Calculations accounted for *active days*, when citizens actively engaged in activities on the platform (i.e., uploading photographs, taking the tutorial, identifying and validating species, analyzing data, posting and commenting in the forum), and *lurking days*, when citizens did not engage in any of these activities but instead browsed through the content of the platform. Days without login and lurking days summed up to the *days elapsed* between two active days. The difference between the very first and the final, last day of login defined the *total days a participant was linked* to the project, while the time between the very first day of login and the day the project ended defined the *potential days a participant could be linked*.

Based on this typology of activities, we calculated engagement metrics in accordance with earlier studies [5,6], which stipulated that participants had to be active for at least two days in order to calculate the engagement metrics. All other participants were only visitors per definition. We established definitions for the following metrics and indicate below the abbreviation for the metric and its score range, if applicable, in brackets:

The ratio of active and total days a participant was linked to the project was the *activity ratio (a* [0; 1]). The ratio of lurking and potential days a participant could be linked to the project was the *lurking ratio (l* [0; 1]). The mean hours of a participant's active contribution during the active days was the *daily devoted time (d* [0; 24]). We also counted the number of days between each of a participant's active days and then calculated the standard deviation of these days in between. Thus, we estimated how regularly a participant actively contributed to the project which was the *variation in periodicity (v)*. Finally, the ratio of total and potential days for a participant in the project was the *relative activity duration (r* [0; 1]).

**Clustering.**    Apart from the question of how participants engaged in the different CS projects, we also investigated whether there were different engagement profiles based on the engagement metrics. Therefore, we performed cluster analyses based on previous reporting guidelines [34]. As the clustering based on engagement metrics was only possible for participants with more than two active days [5,6], we considered $N = 131$ participants for clustering, $n_1 = 121$ participants in the data collection phase, and $n_2 = 36$ participants in the data analysis phase in the first field study. Due to a lower number of participants, we did not conduct a cluster analysis in the second field study. The flow of participants between the emerging clusters of the data collection and the data analysis phase in the first field study was plotted in a Sankey diagram and analyzed for deviances in tests of symmetry [35,36] with further post hoc comparisons [37] in R (v3.6.1).

The clustering procedure was similar to procedures in previous work [5,6]: Prior to clustering, we normalized each engagement metric to an interval of [0; 1] so that all engagement metrics had equal weight in the clustering process [38]. We used SPSS (v24) for cluster analysis and identified engagement profiles by clustering the engagement metrics on the basis of the minimal within-groups sum of squares in each cluster. Subsequent application of hierarchical agglomerative clustering and k-means clustering optimized the results [39]. First, the hierarchical agglomerative clustering algorithm revealed the number of possible clusters. Afterward, the number of possible clusters was used as a specification for the k-means clustering procedure. The centers from hierarchical clustering were the initial centroids in the k-means clustering procedure, to reduce noise and iteration time [39]. As a method for determining the final number of clusters ('stopping rules'), we inspected the dendrograms and the within-sum of squares. Dendrograms displayed the increases in dissimilarity within each cluster by squared Euclidean distances during the merging procedure. The within-groups sum of squares decreased with the number of identified clusters (being zero, when each case is its own cluster; [40]). Hence, the quality of identified clusters was inspected based on the within-groups sum of squares representing homogeneity in the engagement metrics of cluster members. Repeating the clustering for the whole sample of the first field study and two subsamples of the data collection and analysis phases provided validity for the clusters we obtained [34].

## Results

In a first step, we compared both field studies (i.e., the first field study, 'Wildlife Researchers 1', and the second field study, 'Wildlife Researchers 2') with three previously conducted CS projects (i.e., Weather-it: [5]; Milky Way and Galaxy Zoo: [6]). We also compared the data collection and data analysis phases in both 'Wildlife Researchers 1' and 'Wildlife Researchers 2'. In both cases, we based our comparisons on the calculated engagement metrics. Means and standard deviations of all engagement metrics are presented in Table 1.

In a second step, we first calculated engagement profiles through cluster analysis for 'Wildlife Researchers 1'. We next conducted separate cluster analyses for the data collection and data analysis phase in order to differentiate engagement profiles and analyzed the transitions of participants between data collection and data analysis.

### Engagement metrics

We hypothesized that participants would engage for longer periods and more regularly in the two field studies of our collaborative CS project (i.e., 'Wildlife Researchers 1' and 'Wildlife Researchers 2') than participants in other contributory projects (see Hypothesis 1). Hence, we compared the engagement metrics (i.e., the activity ratio *[a]*, daily devoted time *[d]*, relative activity duration *[r]*, variation in periodicity *[v]*, and lurking ratio *[l]*) of 'Wildlife Researchers

**Table 1. Engagement metrics for two field studies of our CS project and three other CS projects (between projects) and, separately, for the data collection and data analysis phases in the two field studies of our CS project (within 'Wildlife Researchers 1', within 'Wildlife Researchers 2').**

| | Between projects | | | | | Within Wildlife Res. 1 | | Within Wildlife Res. 2 | |
|---|---|---|---|---|---|---|---|---|---|
| | Wildlife Res. 1 | Wildlife Res. 2 | Weather-it | Milky Way | Galaxy Zoo | data collect. | data analysis | data collect. | data analysis |
| | $(N_1 = 141)$ | $(N_2 = 132)$ | $(N = 77)$ | $(N = 6,093)$ | $(N = 23,547)$ | $(n = 61)$ | | $(n = 33)$ | |
| | $M$ $(SD)$ | $M$ $(SD)$ | $M$ $(SD)$ | $M$ $(SD)$ | $M$ $(SD)$ | $M_1$ $(SD_1)$ | $M_2$ $(SD_2)$ | $M_1$ $(SD_1)$ | $M_2$ $(SD_2)$ |
| $a$ | .20 (.12) | .18 (.15) | .32*** (.35) | .40*** (.40) | .33*** (.38) | .36*** (.19) | .16 (.10) | .31*** (.19) | .12 (.07) |
| $d$ | 0.78 (0.60) | 0.76 (0.50) | — | 0.44*** (0.54) | 0.32*** (0.40) | 0.82 (0.56) | 0.70 (0.53) | 0.77*** (0.31) | 0.44 (0.37) |
| $r$ | .67 (.20) | .71 (.28) | .43*** (.44) | .20*** (.30) | .23*** (.29) | .85*** (.12) | .59 (.22) | .89** (.13) | .80 (.13) |
| $v$ | 5.32 (4.33) | 5.26 (4.05) | 5.11 (5.36) | 18.27*** (43.31) | 25.23*** (49.16) | 2.33 (1.92) | 5.15** (4.76) | 1.86 (1.18) | 3.98+ (3.73) |
| $l$ | .20 (.16) | .26 (.18) | .35*** (.39) | — | — | .11 (.15) | .31*** (.21) | .27 (.15) | .46*** (.24) |

$a$ = activity ratio; $d$ = daily devoted time; $r$ = relative activity duration; $v$ = variation in periodicity; $l$ = lurking ratio.

$p$-values refer to the between projects comparison with 'Wildlife Researchers 1'. $+ p < .1$ $** p < .01$; $*** p < .001$.

1' and 'Wildlife Researchers 2' with the engagement metrics of three previously conducted CS projects (i.e., Weather-it: [5]; Milky Way and Galaxy Zoo: [6]). To accomplish this comparison, we calculated t-tests, effect sizes (Cohens' $d$), and 95% confidence intervals.

**Comparison with other projects.** The analyses for the category of *relative activity duration (r)* revealed that participants stayed more days in the online community of 'Wildlife Researchers 1' and 'Wildlife Researchers 2' than participants in the CS project Weather-it, $t_{\text{wildlife 1}}(140) = 14.623, p < .001, d_r = 0.78$, 95% CI [0.50; 1.07], $t_{\text{wildlife 2}}(131) = 11.560, p < .001, d_r = 0.81$, 95% CI [0.54; 1.10]; than participants in the project Milky Way, $t_{\text{wildlife 1}}(140) = 28.498, p < .001, d_r = 1.58$, 95% CI [1.41; 1.75], $t_{\text{wildlife 2}}(131) = 21.143, p < .001, d_r = 1.70$, 95% CI [1.53; 1.88]; and than participants in the project Galaxy Zoo, $t_{\text{wildlife 1}}(140) = 26.688, p < .001, d_r = 1.52$, 95% CI [1.35; 1.69], $t_{\text{wildlife 2}}(131) = 19.893, p < .001, d_r = 1.66$, 95% CI [1.49; 1.83] (see Table 1). These findings supported Hypothesis 1.

With reference to the *variation in periodicity*, participants contributed more regularly *(v)* in 'Wildlife Researchers 1' and 'Wildlife Researchers 2' than participants in the project Milky Way, $t_{\text{wildlife 1}}(130) = -34.262, p < .001, d_v = -0.30$, 95% CI [−0.48; −0.13], $t_{\text{wildlife 2}}(120) = -35.323, p < .001, d_v = -0.30$, 95% CI [−0.48; −0.12], and than participants in the project Galaxy Zoo, $t_{\text{wildlife 1}}(130) = -52.678, p < .001, d_v = -0.41$, 95% CI [−0.58; −0.23], $t_{\text{wildlife 2}}(120) = -54.225, p < .001, d_v = -0.41$, 95% CI [−0.59; −0.23]. Yet, 'Wildlife Researchers 1' and 'Wildlife Researchers 2' did not differ from the project Weather-it, $t_{\text{wildlife 1}}(130) = 0.561, p = .576, t_{\text{wildlife 2}}(120) = 0.417, p = .677$ (see Table 1). These results supported Hypothesis 1.

Participants *devoted more time on active days (d)* within the online community of 'Wildlife Researchers 1' and 'Wildlife Researchers 2' than participants in the project Milky Way, $t_{\text{wildlife 1}}(140) = 6.707, p < .001, d_d = 0.63$, 95% CI [0.46; 0.80], $t_{\text{wildlife 2}}(131) = 7.464, p < .001, d_d = 0.59$, 95% CI [0.42; 0.77], and than participants in the project Galaxy Zoo, $t_{\text{wildlife 1}}(140) = 9.102, p < .001, d_d = 1.15$, 95% CI [0.98; 1.31], $t_{\text{wildlife 2}}(131) = 10.235, p < .001, d_d = 1.10$, 95% CI [0.93; 1.27] (see Table 1). There was no data available on daily devoted time for the project Weather-it.

However, participants in the online community of 'Wildlife Researchers 1' and 'Wildlife Researchers 2' showed a lower *activity ratio (a)* than participants in the project Weather-it, $t_{\text{wildlife 1}}(140) = -11.934, p < .001, d_a = -0.52$, 95% CI [−0.81; −0.24], $t_{\text{wildlife 2}}(131) = -10.707, p < .001, d_a = -0.58$, 95% CI [−0.86; −0.29], than participants in the project Milky Way, $t_{\text{wildlife 1}}(140) = -19.682, p < .001, d_a = -0.51$, 95% CI [−0.67; −0.34], $t_{\text{wildlife 2}}(131) = -16.852, p < .001, d_a = -0.56$, 95% CI [−0.73; −0.38], and than participants in the project Galaxy Zoo, $t_{\text{wildlife}}$

$_1(140) = -12.902$, $p < .001$, $d_a = -0.34$, 95% CI [−0.51; −0.18], $t_{\text{wildlife } 2}(131) = -11.475$, $p < .001$, $d_a = -0.40$, 95% CI [−0.57; −0.23] (see Table 1).

Finally, participants in the online community of 'Wildlife Researchers 1' and 'Wildlife Researchers 2' *lurked (l)* less on the online platform than participants in the project Weather-it, $t_{\text{wildlife } 1}(140) = -11.154$, $p < .001$, $d_l = -0.57$, 95% CI [−0.85; −0.28], $t_{\text{wildlife } 2}(131) = -5.866$, $p < .001$, $d_l = -0.33$, 95% CI [−0.61; −0.04] (see Table 1). There was no data available for the lurking ratio in the Milky Way and Galaxy Zoo projects.

**Comparisons of phases within the project.** We also hypothesized that participants in the two field studies of our CS project (i.e., 'Wildlife Researchers 1' and 'Wildlife Researchers 2') would engage more actively in the data collection phase than in the data analysis phase (see Hypothesis 2). To test this hypothesis, we compared the engagement metrics of the data collection and the data analysis phases within both field studies. For this comparison, we again calculated t-tests, effect sizes (Cohens' *d)*, and 95% confidence intervals. The analyses showed that participants' *activity ratio (a)* was higher for the data collection than for the data analysis phase in both field studies, $t_{\text{wildlife } 1}(60) = 7.27$, $p < .001$, $d_{a1,2} = 0.76$, 95% CI [0.39; 1.13], $t_{\text{wildlife } 2}(32) = 6.41$, $p < .001$, $d_{a1,2} = 0.92$, 95% CI [0.42; 1.43]. The *lurking ratio (l)* was higher for the data analysis than for the data collection phase in both field studies, $t_{\text{wildlife } 1}(60) = -7.41$, $p < .001$, $d_{l1,2} = -0.95$, 95% CI [−1.32; −0.57], $t_{\text{wildlife } 2}(32) = -4.51$, $p < .001$, $d_{l1,2} = -1.08$, 95% CI [−1.60; −0.57] (see Table 1). These results supported Hypothesis 2. In the category of *relative activity duration (r)*, participants stayed linked to the data collection phase for a longer period of time than to the data analysis phase in both field studies, $t_{\text{wildlife } 1}(60) = 8.87$, $p < .001$, $d_{r1,2} = 1.69$, 95% CI [1.28; 2.10], $t_{\text{wildlife } 2}(32) = 3.04$, $p = .005$, $d_{r1,2} = 0.55$, 95% CI [0.05; 1.05]. In terms of the *variation in periodicity (v)*, participants contributed more regularly during the data collection than during the data analysis phase in both field studies, $t_{\text{wildlife } 1}(32) = -3.74$, $p = .001$, $d_{v1,2} = -1.36$, 95% CI [−1.89; −0.82], $t_{\text{wildlife } 2}(11) = -1.86$, $p = .090$, $d_{v1,2} = -1.29$, 95% CI [−2.17; −0.41]. Finally, participants' *daily devoted time (d)* was significantly higher for the data collection than for the data analysis phase in 'Wildlife Researchers 2', $t_{\text{wildlife } 2}(32) = 4.30$, $p < .001$, $d_{d1,2} = 0.82$, 95% CI [0.32; 1.32], but not in 'Wildlife Researchers 1', $t_{\text{wildlife } 1}(60) = 1.43$, $p = .159$, $d_{d1,2} = 0.18$, 95% CI [−0.18; 0.54] (see Table 1). These findings were partly in line with Hypothesis 2.

## Engagement profiles

In the next step, we identified engagement profiles based on participants' engagement metrics in 'Wildlife Researchers 1' with a cluster analysis for $N = 131$ participants (i.e., 93%) with more than two active days. This number of participants is much higher than in other projects where less than half of the participants had more than two active days [5]. We can therefore conclude that our participants were active and engaged.

We exploratively investigated whether the engagement profiles in our CS project were similar to other collaborative projects such as the project 'Weather-it' (see also Hypothesis 1). The optimal number of clusters from the clustering procedure was estimated by its dendrogram and cross-validated by the within-groups sum of squares, which is interpreted as a scree-plot in factor analysis. The inspection of the dendrogram for 'Wildlife Researchers 1' indicated two, three, four, five or eight possible clusters, when the dendrogram was broken at five cut-off values of the Euclidean distances ('20', '14', '11', '9', '4') and the intersections between horizontal and vertical lines were counted (see S1 File). For cross-validation of the possible clusters, we plotted the change in slope of the within-groups sum of squares: Inconsistencies in the increase for the step from five to four clusters indicated that dissimilar clusters had been merged. Hence, five clusters were the minimal number of clusters (see S2 File). Following this

reasoning, we identified five different engagement profiles which were similar to previously identified engagement profiles. This was done with respect to the kinds of profiles, but not with respect to the absolute values of the engagement metrics. Fig 2 represents the standardized engagement metrics for each identified cluster.

The most prevalent profile was the *loyal (lo)* engagement profile ($n_{lo}$ = 62). Participants with this profile stayed almost until the end of the project (high relative activity duration, $r$ = 0.79), returned regularly to the community (low variation in periodicity, $v$ = 0.19), and actively contributed to the community at least one day out of five days (moderate activity ratio, $a$ = 0.23, see Fig 2, A1).

Participants in the *hardworking (ha)* engagement profile ($n_{ha}$ = 5) outperformed other participants in their activity ratio and daily devoted time. They spent almost 2 hours and 50 minutes during an active day on the platform (high daily devoted time, $d$ = 0.78) and contributed to the CS project every third day (high activity ratio, $a$ = 0.29). However, their relative activity duration within the project was shorter ($r$ = 0.58) than that of participants in the loyal profile (see Fig 2, A2), indicating that participants in this hardworking profile left the project earlier after the work was done.

The *persistent (pe)* engagement profile ($n_{pe}$ = 7) was characterized by the highest variation in periodicity ($v$ = 0.79), as participants with this profile participated irregularly. Although persistent participants visited the platform only sporadically, they stayed connected to the project almost until the end of the project, like the participants with a loyal engagement profile (high relative activity duration, $r$ = 0.80). Their very low activity ratio ($a$ = 0.04) indicated low contributions to the CS project (see Fig 2, A3).

Participants in the *lurker (lu)* engagement profile ($n_{lu}$ = 28) were connected to the project for a long period of time (high relative activity duration, $r$ = 0.78). Yet, on more than half of their visiting days, they viewed camera trap pictures or read discussions in the forum (high lurking ratio, $l$ = 0.57) without actively contributing to the CS project (low activity ratio, $a$ = 0.11, see Fig 2, A4).

Participants in the *moderate (mo)* engagement profile ($n_{mo}$ = 29) had a balanced engagement across all engagement metrics with no defining key characteristic. They participated for a shorter amount of time (moderate relative activity duration, $r$ = 0.43) at a moderate to high activity level (moderate to high activity ratio, $a$ = 0.25, see Fig 2, A5). However, the number of participants in this engagement profile was as high as the number of participants in the lurker engagement profile.

**Engagement profiles for data collection and data analysis.**   Besides the identification of engagement profiles across the whole project, we further analyzed how engagement profiles differed and changed between the data collection and data analysis phases. Hence, we conducted separate cluster analyses based on engagement metrics of participants with more than two active days for the data collection ($n_1$ = 121) and data analysis phase ($n_2$ = 36). We exploratively investigated whether engagement profiles differed between the data collection and data analysis phases (see also Hypothesis 2). The sheer numbers of participants already indicated that in the data collection phase, many more participants had more than two active days than participants in the data analysis phase.

Again, we inspected the dendrograms and the within-groups sum of squares in both clustering procedures for the data collection and the data analysis phases. The inspection of the dendrogram revealed two to seven or nine possible clusters for the *data collection phase*. Cross-validation by the within-groups sum of squares showed that within-cluster dissimilarity was still small for six clusters. For the *data analysis phase*, two to six possible clusters were identified. Cross-validation by the within-groups sum of squares indicated that within-cluster dissimilarity was still small for six clusters (see S1 and S2 Files).

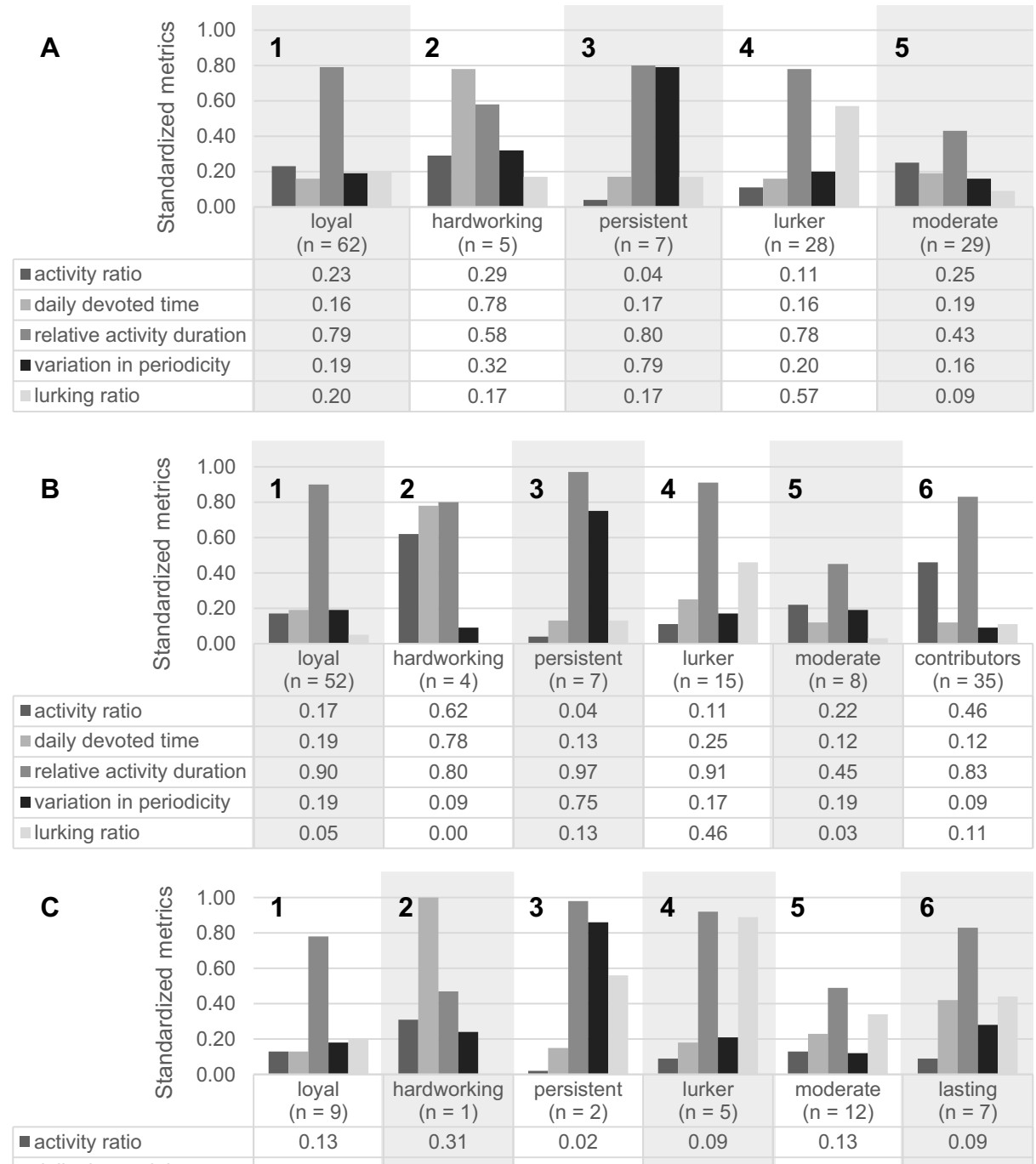

**Fig 2. Engagement profiles for participants.** Engagement profiles (A) within the 'Wildlife Researchers 1' project ($N = 131$) and the two phases of (B) data collection ($n_1 = 121$) and (C) data analysis ($n_2 = 36$) for participants with more than two active days are based on the standardized engagement metrics for clustering [0; 1].

For the *data collection* phase, we identified the same five engagement profiles as for the whole project. Participants in the loyal engagement profile ($n_{lo1} = 52$) were well connected to the CS project (high relative activity duration, $r = 0.90$) and returned regularly (low variation in periodicity, $v = 0.19$, see Fig 2, B1). Participants with a hardworking engagement profile ($n_{ha1} = 4$) devoted a lot of time (high daily devoted time, $d_{ha1} = 0.78$) on many active days (high activity ration, $a_{ha1} = 0.62$) and stayed well connected to the CS project (high relative activity duration, $r = 0.80$, see Fig 2, B2). Participants in the persistent engagement profile ($n_{pe1} = 7$) participated irregularly (high variation in periodicity, $v = 0.75$) but were still well connected to the CS project (high relative activity duration, $r = 0.97$, see Fig 2, B3). Participants in the lurking engagement profile ($n_{lu1} = 15$) were also well connected to the project (high relative activity duration, $r = 0.91$) but mostly browsed the content of the platform passively (high lurking ratio, $l = 0.46$, see Fig 2, B4). Participants in the moderate engagement profile ($n_{mo1} = 8$) were moderately connected to the CS project (moderate relative activity duration, $r = 0.45$) and also contributed actively on a moderate level (moderate activity ratio, $a = 0.22$, see Fig 2, B5).

Next to these five engagement profiles, we identified a sixth engagement profile which was characteristic for the data collection phase: Participants with a *contributor (co)* engagement profile ($n_{co1} = 35$) were especially active during the upload, identification, and validation of camera trap pictures (high activity ratio, $a = 0.46$). Although they did not devote much time on active days (low daily devoted time, $d = 0.12$), they were well connected to the CS project (high relative activity duration, $r = 0.83$, see Fig 2, B6). Besides the loyal participants, participants with the contributor engagement profile were the most common group during the data collection phase.

For the *data analysis* phase, we also identified the same engagement profiles as for the whole project. Participants in the loyal engagement profile ($n_{lo2} = 9$) still stayed well connected during the data analysis phase (high relative activity duration, $r = 0.78$), but used some of their time for passively browsing the content of the platform (moderate lurking ratio, $l = 0.20$, see Fig 2, C1). The participant in the hardworking engagement profile ($n_{ha2} = 1$) devoted a lot of time to data analysis (high daily devoted time, $d = 1.00$) and actively contributed (high activity ratio, $a = 0.31$, see Fig 2, C2). Participants in the persistent engagement profile ($n_{pe2} = 2$) were still well connected to the CS project and the data analysis phase (high relative activity duration, $r = 0.98$). But they visited the platform irregularly (high variation in periodicity, $v = 0.86$) and used their time on the platform for browsing (high lurking ratio, $l = 0.56$, see Fig 2, C3). Participants in the lurking engagement profile ($n_{lu2} = 5$) mostly browsed the content of the platform without actively contributing (high lurking ratio, $l = 0.89$) but were still well connected to the platform and the CS project (high relative activity duration, $r = 0.92$, see Fig 2, C4). Finally, participants in the moderate engagement profile ($n_{mo2} = 12$) were moderately connected to the CS project (moderate relative activity duration, $r = 0.49$), returned regularly (low variation in periodicity, $v = 0.12$), and mostly used their time for passively browsing the content of the platform (moderate lurking ratio, $l = 0.34$, see Fig 2, C5).

In addition to these five profiles, we identified a sixth engagement profile characteristic of the data analysis phase: Participants with a *lasting (la)* engagement profile ($n_{la1} = 7$) stayed for a relatively long time (high relative activity duration, $r = 0.83$) during the data analysis phase and, hence, extended their participation in the project. They devoted a lot of time per day to the CS project (high daily devoted time, $d = 0.42$), yet mostly for browsing content on the platform (moderate lurking ratio, $l = 0.44$). They also returned irregularly (moderate variation in periodicity, $v = 0.28$, see Fig 2, C6). Overall, the number of participants within every profile in the data analysis phase was much lower than the number of participants in the data collection phase. Although the five engagement profiles that we found across the field study were present

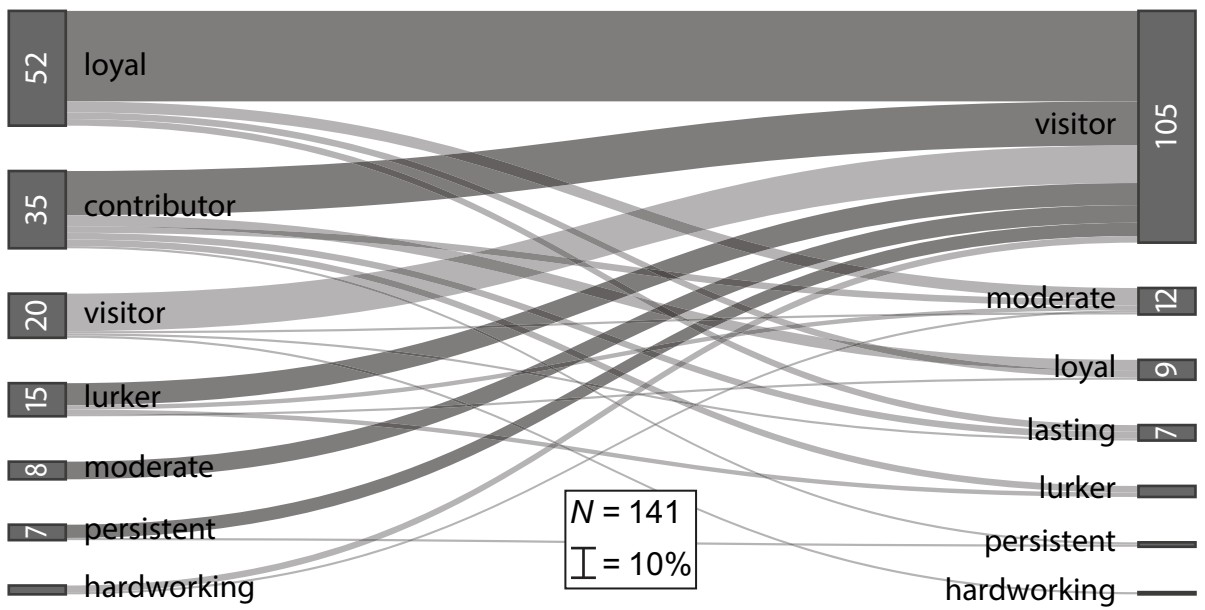

**Fig 3. Sankey diagram of participants' transitions.** Transitions occur between engagement profiles ($N$ = 141; 'Wildlife Researchers 1') between the data collection (left) and the data analysis phase (right), with all $p$s < .05 (dark grey) [Interactive version online in S3 File].

for both the data collection and data analysis phase, the sixth profiles in each phase differed. Together with the decrease in the number of participants, these differences indicated that the data collection and data analysis phases produced qualitatively and quantitatively different engagement profiles.

**Transitions between engagement profiles.** Lastly, we analyzed the transitions of participants between the engagement profiles from the data collection towards the data analysis phase. To do this, we plotted the number of participants in the engagement profiles for both phases in a Sankey diagram (see Fig 3). Then, we compared the distribution of participants in the engagement profiles between the data collection and data analysis phase in a nominal symmetry test with $X^2$ statistics. Transitions of participants between the engagement profiles in the data collection and data analysis phases were tested by further post hoc tests, which provided $p$-values but no test statistics [37]. We exploratively investigated whether the participant numbers in the engagement profiles differed for the data collection and data analysis phase, as data analysis allowed for higher levels of participation, but also required higher levels of cognitive engagement. We based these considerations on previous findings regarding participants' reasons to participate in CS projects, and on the number of participants with more than two active days in the data collection and data analysis phases.

The engagement profiles of the data collection and data analysis phases each had different numbers of participants allocated to them, $X^2$ = 75.34, $p$ < .001. Overall, the majority of participants ($n$ = 88) changed their profile from data collection to data analysis and in most cases adopted a visitor engagement profile ($n$ = 105 participants with less than two active days were visitors per definition). To be more specific, the most prevalent engagement profile during data collection was the *loyal* engagement profile, with 52 participants. However, from these participants, 41 participants became visitors during the data analysis phase, $p$ < .001, as they reduced their active contribution to less than two active days. Only 5 participants continued their engagement in the moderate profile, $p$ = .063. Thus, there was a high dropout rate within the loyal engagement profile, and loyal contribution was limited to the data collection phase.

The 35 participants in the *contributor* engagement profile actively participated almost half of all the days during data collection. Of these participants, 20 participants dropped out of this profile during the data analysis phase, $p < .001$, and turned into visitors with no more than two active days. The other contributing participants stayed active for more than two days during the data analysis phase and were distributed across different engagement profiles, with the majority participating almost until the end (i.e., loyal engagement: $n_{co1;lo2} = 5$), $p = .063$.

With respect to the other profiles, the picture was the same, only the numbers were smaller. Of the 7 participants in the *persistent* engagement profile in the data collection phase, 6 participants became visitors in the data analysis phase, $p = .031$. Of the 15 participants in the *lurking* engagement profile in the data collection phase, 10 participants became visitors in the data analysis phase, $p = .002$. Finally, all 8 participants in the *moderate* engagement profile in the data collection phase were also visitors during the data analysis phase, $p = .039$.

To sum up, the results of post hoc comparisons for each engagement profile indicated that participants with actively contributing profiles (i.e., loyal and contributing) decreased their activities after the data collection phase and participated in more passive engagement profiles. Nevertheless, participants in the contributing and the loyal engagement profiles were still more likely to actively contribute in the data analysis phase than participants in any other profile.

## Discussion

The increasing popularity of CS has led to the development of a vast number of projects engaging thousands of citizens. In this respect, it is important to know how participants actually participate in CS projects in order to provide relevant insights for researchers and practitioners in CS to aid them in preparing and conducting future CS projects. To provide such insights, we analyzed participants' actual engagement patterns in two field studies of a collaborative wildlife urban ecology CS project, compared them with other contributory and collaborative CS projects, and tested whether participants' engagement differed for the two phases of the field studies (i.e., data collection and data analysis). The findings of our research indicate that participants were engaged for longer periods of time, returned more frequently, and devoted more time to our collaborative project than participants in more contributory projects. Our findings also revealed that participants contributed more actively during the data collection than during the data analysis phase. This difference was similarly apparent in participants' transitions from being loyal or contributing participants during data collection to being mostly lurking or visiting participants during data analysis. Thus, this research provides evidence that collaborative CS projects enhance participants' commitment to the project, but that it is especially data collection that promotes the most active contribution.

### Engagement metrics in collaborative CS online communities

First, our results correspond to previous findings regarding collaborative projects (i.e., Weather-it). In collaborative projects, participants often stay for a longer period of time, contribute more regularly, and devote more daily hours to the research activities in the online community than in contributory projects. This is the case because collaborative CS projects open the research process to facilitate participants' own investigations [5]. In contributory CS projects (i.e., Milky Way and Galaxy Zoo: [2,6]), however, participants' contributions are limited to collecting and processing data for others (i.e., the scientists). These projects often struggle with sporadic engagement of participants. In the case of our research presented here, providing participants with the opportunity to perform tasks beyond collecting data seemed to increase their allocated time (i.e., daily devoted time) over a longer period of time (i.e., relative

activity duration) on a more regular basis (i.e., variation of periodicity), while their active engagement (i.e., activity ratio) did not seem to increase in general. These findings are in line with previous research, which has suggested that offering participants a choice of tasks in CS projects encourages participant contribution [2,25].

## Engagement metrics differ between data collection and data analysis

Second, our results expand upon previous research studies on collaborative CS projects regarding participants' engagement in different phases of the project [5,7]. Our results demonstrated that participants' active contribution (i.e., activity ratio) was much higher in the data collection phase than in the data analysis phase, while the reverse was true for their passive behavior (i.e., lurking ratio). Hence, our results provide indications that the lower activity ratio that was often found in collaborative CS projects [5,6] may have resulted from the fact that participants' active contributions were skewed towards data collection. Based on these results, our study disentangled confounds in previous research on collaborative CS projects [5]. Opening the research process to collaborative CS projects did not enhance any active engagement besides the higher commitment of participants. Contrary to the intentions of collaborative CS projects [19], a differentiated analysis of the activities for data collection and data analysis showed that participants' engagement did not correspond to the goal of higher levels of participation [9,41,42].

## Participants' engagement profiles for data collection and data analysis

Third, our research corresponds to previous findings on engagement profiles in contributory and collaborative CS projects. The clustering of engagement metrics in our CS project revealed five profiles that were also found in earlier studies (i.e., the loyal, hardworking, persistent, lurker, and moderate engagement profiles; 5,6). Besides these five profiles, we found two additional engagement profiles when separately clustering the engagement metrics for the data collection and data analysis phases. The *contributor* engagement profile was exclusive to the data collection phase and reflected the high activity of participants in uploading, identifying, and validating the camera trap photographs. The *lasting* engagement profile [6] was only present in the data analysis phase, representing participants who contributed irregularly but remained connected to the project. Thus, together with the contributor profile, the engagement profiles in the data collection phase were characterized by high numbers of participants and active contributions. In contrast, together with the lasting profile, the engagement profiles in the data analysis phase were characterized by low numbers of participants and more passive contributions.

Previous research has indicated that designing CS projects for higher levels of participation in the research process [19] does not necessarily lead to higher engagement [9,41,42]. This is in line with our findings, which may show that participants' motivation and knowledge constrained their engagement when dealing with data analysis tasks (e.g., doing statistics: [9]; cf. [43]). These previous and our current findings provide evidence that participants' contributions were much more frequent and more active in the data collection than in the data analysis phase. One explanation might be that participants in CS projects find their knowledge insufficient for gaining evidence from data. Therefore, they may not see the engagement in data analysis and dealing with research evidence as realistic behavioral actions [44].

## Supporting the transition from collecting data to analyzing data

Last, our results for the transition of participants from the different engagement profiles in the data collection and the data analysis phases both correspond to previous findings on

participants' motivation to engage [9,42] and simultaneously shed new light on the behavioral engagement patterns in CS projects [5,6]. The engagement profiles in the two phases differed qualitatively (i.e., characteristics of the profiles) and quantitatively (i.e., number of participants in the profiles). Many more participants actively contributed during data collection (i.e., in the loyal, hardworking, and contributor profiles) than during data analysis (i.e., in the visitor and lasting profiles). This finding may be evidence that some participants were probably fine with collecting large data sets for scientists [2,9]. Despite the low activity during data analysis, those participants who actively engaged on a high level in data collection (i.e., participants in the contributor and loyal profiles) were more likely to also engage in the data analysis phase. We suppose as a post hoc explanation for why highly active participants also engaged in data analysis that those participants felt ownership (i.e., an individuals' feeling of possessing a concrete or abstract object or entity; [45]) for the data, that they themselves collected and contributed. This could well be the case, as research on project ownership has indicated that high project ownership promotes deeper involvement in problem solving in the inquiry process [46]. Other possible explanations for why highly active participants in the data collection also engaged in data analysis could be that those participants were in general more motivated to participate in all phases [47] or even held a specific set of knowledge and skills that helped them to analyze the data as well [48].

## Implications

For online communities in CS projects, our results have theoretical, methodological, and practical implications. By expanding on previous theoretical considerations ([4,19]; cf. [9]), our results suggest that simply offering opportunities for higher levels of participation in CS may not correspond to participants' preferred level of engagement with research tasks such as data analyses. Although participants' commitment to collaborative CS projects offering deeper involvement was higher than in contributory projects [5], further research is necessary to find ways to engage participants in higher-level tasks more actively. Such research could, for instance, define the boundary conditions of engagement more clearly [9] and offer trial activities (such as tutorials) that motivate and facilitate participants' willingness to get involved in cognitive tasks of higher order [25].

Our analyses also revealed differing engagement metrics and profiles across the two phases of the CS projects. These findings point to a possible methodological confound in previous research when tracking participants' behavior in online CS communities [5,6]. For an unbiased comparison of behavioral engagement patterns between contributory and collaborative CS projects, the actual engagement has to be tracked in, for example, log files. Moreover, close care should be taken that the engagement patterns are tracked separately for the different tasks. Only in this way will a differentiated picture of actual engagement in CS projects be possible.

Regarding practical implications for online communities in CS projects, supporting citizens with scaffolding tools for data analysis (i.e., digital tools that facilitate data inspection through visualizations and automated testing procedures) do not seem to suffice. Previous suggestions on the design of online platforms for CS [25] need to be refined on the basis of participants' individual motives and their resulting engagement. Mechanisms to support specific engagement profiles [5] need to consider participants' engagement depending on the different tasks of data collection and analysis. Post hoc, we suggest that how citizens' perceive their roles (i.e., as data collectors) may hinder them from taking on a new role requiring cognitive involvement with data analyses. Hence, besides the support with scaffolding tools within the inquiry process, citizens may also need encouragement to change their role from data collector to data inquirer.

Targeting the most active participants during data collection for encouragement could also facilitate the transition of participants to data analysis. Citizens who have already contributed with high engagement during data collection may possibly be more likely to engage with data analysis, as they have already invested a lot of time and effort which in turn may increase their project ownership. Further research should investigate in detail whether participants' already high investment as a data collector may be an important lever to facilitate their transition from the data collector to the data inquirer role (see also [49]).

## Strength, limitations, and future research

This research used two field studies of an existing collaborative CS project involving a sample of citizens representative of CS projects [43]. With the tracking data of the log files, we provided data on citizens' actual engagement behavior in the online community of the CS project. Thus, this research presents important findings of actual engagement patterns in collaborative CS projects that previous research has only sparsely investigated and that is relevant for the CS community.

At the same time, we need to discuss some limitations of our research. Effects of the chronological order of data collection and analysis and their temporal restriction have potentially limited our findings. Participants first had to collect data on wildlife before they were able to analyze it. Although the order is valid from a scientific point of view (i.e., data has to be collected before being analyzed), analyses of contribution patterns in CS projects indicate that participants' activities rapidly decrease over time [2]. Hence, future research should examine the effects of chronological order on participants' engagement by providing opportunities for analyzing data from the beginning of the project. One possibility as an example would be to provide data from previous CS projects. If participants take advantage of such an opportunity, a mere change in the chronological order of data collection and analysis in earlier stages could support the change of roles from data collectors to data inquirers in CS projects. Furthermore, engagement with data collection and analysis was time-restricted, so that more resource availability could have motivated more participant engagement. Even though comparable CS projects limited the facilitation of participants' engagement in time [7], future studies should investigate the availability of resources as an influencing factor in CS.

A second limitation concerns the sample size in our field studies. Especially, the sample sizes for the engagement profiles in the data analysis phase were smaller than the sample sizes of those profiles in the data collection phase. While this difference may limit the conclusion about participants' engagement in the data analysis, the quantitative and qualitative changes between the profiles also highlight the differences in engagement between the data collection and the data analysis phase. In general, the number of participants was limited to the camera traps available in the CS project. Therefore, we did not estimate the sample size a priori to account for the frequent dropout of participants in CS projects. Hence, due to the lower number of participants in the second field study, we were unable to perform further cluster analysis. Payment of the participants could, on the one hand, have assured their continued participation. On the other hand, participants' voluntary participation allowed us to observe engagement patterns unbiased in terms of monetary reward or social desirability, because many CS projects require such voluntary engagement of citizens (e.g., White House memo on CS; [50]). In future research on online CS communities, the sample size could be increased by adopting research methodology to include participants' own data collection tools or by paying the participants.

Although the suggested engagement metrics [5,6] were useful to track participants' engagement and led to comparable results, they remain on the level of quantifying behavioral engagement. To go a step further, the quality of participants' activity in terms of relevant

contributions to the community's goals should be accounted for as well [32]. Our analysis of different scientific tasks in the online CS community qualified participants' behavior in terms of levels of participation [19]. However, further research should consider the motivation behind participants' contributions. Therefore, in our future research, we would expand our own analyses to include participants' motivations.

## Conclusion

In sum, this research provides evidence that the degree of participants' engagement in the online community of our collaborative CS project depended on the scientific tasks of data collection and data analysis. Our results demonstrated that participants' engagement did not necessarily reflect the intended level of participation in scientific research, but corresponded to previously identified motivations of participants to engage in CS projects. Hence, simply increasing the opportunities for participation in scientific tasks from contributory towards collaborative CS projects does not necessarily lead to enhanced engagement. Future research should explore which design factors in CS projects would in fact facilitate participants' transition from data collectors to data inquirers, from more straightforward to more complicated tasks in terms of cognitive involvement.

## Supporting information

**S1 File.**
(PDF)

**S2 File.**
(PDF)

**S3 File.**
(PDF)

**S1 Dataset. Engagement metrics of 'Wildlife Researchers 1'.**
(SAV)

**S2 Dataset. Engagement metrics of 'Wildlife Researchers 2'.**
(SAV)

**S3 Dataset. Engagement profiles of 'Wildlife Researchers 1'.**
(SAV)

## Author Contributions

**Conceptualization:** Till Bruckermann, Hannah Greving, Milena Stillfried, Anke Schumann, Miriam Brandt, Ute Harms.

**Formal analysis:** Till Bruckermann.

**Funding acquisition:** Miriam Brandt, Ute Harms.

**Investigation:** Till Bruckermann, Hannah Greving, Milena Stillfried, Anke Schumann, Miriam Brandt.

**Methodology:** Till Bruckermann, Hannah Greving, Ute Harms.

**Writing – original draft:** Till Bruckermann, Hannah Greving.

**Writing – review & editing:** Milena Stillfried, Anke Schumann, Miriam Brandt, Ute Harms.

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
