## [Decision Letter · Decision Letter 0]

2 Sep 2022

PONE-D-22-09613

I’m fine with collecting data: Engagement profiles differ depending on scientific activities in an online community of a citizen science project

PLOS ONE

Dear Dr. Bruckermann,

Thank you for submitting your manuscript to PLOS ONE. After careful consideration, we feel that it has merit but does not fully meet PLOS ONE’s publication criteria as it currently stands. Therefore, we invite you to submit a revised version of the manuscript that addresses the points raised during the review process.

The manuscript has been evaluated by three reviewers, and their comments are available below.

The reviewers have raised a number of concerns that need attention. They request additional information and discussion on how citizen engagement profiles can differ in different types of research to provide context to interpret the results with. They also request additional clarity on the definitions of data collection and data analysis, and note concerns with the small sample sizes for the data analysis profiles and that not all claims may be supported. 

Could you please revise the manuscript to carefully address the concerns raised?

We look forward to receiving your revised manuscript.

Kind regards,

Alice Coles-Aldridge

Editorial Office

PLOS ONE

Reviewers' comments:

Reviewer's Responses to Questions

**Comments to the Author**

1. Is the manuscript technically sound, and do the data support the conclusions?

Reviewer #1: Yes

Reviewer #2: Partly

Reviewer #3: Yes

2. Has the statistical analysis been performed appropriately and rigorously? 

Reviewer #1: Yes

Reviewer #2: Yes

Reviewer #3: I Don't Know

3. Have the authors made all data underlying the findings in their manuscript fully available?

Reviewer #1: Yes

Reviewer #2: Yes

Reviewer #3: Yes

4. Is the manuscript presented in an intelligible fashion and written in standard English?

Reviewer #1: Yes

Reviewer #2: Yes

Reviewer #3: Yes

5. Review Comments to the Author

Reviewer #1: Thank you for exploring the extent to which citizens are engaged in different tasks of the scientific process. Many times, citizen involvement in research is based around tokenism, and does not result in meaningful participation of citizens in the work.

Revision suggestions:

Adding some discussion (2-3 lines, maybe a paragraph) around how citizen engagement profiles can differ in different types of research (e.g., health sciences vs. zoological sciences etc) in the background could provide the reader with some context to interpret the results with.

Reviewer #2: The paper builds on an important issue under examination in the citizen science field-- how to deepen participant-engagement to benefit the citizens and the science. One concern I have is the authors' definition of "data collecting" . Here, it is described as the act of setting out camera traps and identifying/labeling the images captured by the cameras. That actually could be considered data analysis. They are analyzing the data collected by camera traps. The act of making and sharing the observations (by going into nature to take and upload the pictures, for example), is what I expected to see in this paper. Perhaps the authors can address this in the paper a bit? There are thousands of citizen science projects that rely on people to collect and share data and typically that means that the participants make and share observations...not retrieve, analyze and label the data. Again, that seems to be a form of data analysis.

Reviewer #3: Thank you for conducting this work examining engagement patterns across projects. In general, I think the paper is very strong and is need of minor revisions.

This statement is unclear, please explain: The standard deviation of days between pairs of sequential active days was the variation in periodicity (v).

The sample sizes for the data analysis profiles are so small - can we really say anything about these profiles?

Unsure that this claim is warranted: "We suppose as a post-hoc explanation that ownership (i.e., an individuals’ feeling of possessing a concrete or abstract object or entity; [38]) for the data, that they themselves collected and contributed, might explain why highly active participants also engaged in data analysis." There may be many other factors at play besides ownership. It would be important for you to include alternate explanations (e.g., not interested/motivated to do analysis, feeling "uncredentialed," lack of educational support for analytical skills.

6. PLOS authors have the option to publish the peer review history of their article (what does this mean?). If published, this will include your full peer review and any attached files.

Reviewer #1: No

Reviewer #2: No

Reviewer #3: No

---

## [Author Response · Author response to Decision Letter 0]

8 Sep 2022

Dear Alice Coles-Aldridge, dear Reviewers,

thank you very much for your positive and helpful feedback and for investing your time and effort in reviewing our manuscript. We appreciate your constructive and valuable feedback that has helped us very much in improving the paper on its way to publication.

We have considered all of your suggestions, and we are convinced that we have made appropriate improvements. In this letter, we address all of your comments from the reviews and explain our revisions in detail. To ensure the transparency of our revisions, we have responded to each comment separately and explain our responses to each comment below. In some cases, we have also pasted the new or changed paragraphs in quotation marks and indented them. For a full account of our changes, please refer to the revised manuscript in which we have highlighted the changes in blue color.

Comments from the Editor and Reviewers:

Editor:

Comment 1:

Thank you for submitting your manuscript to PLOS ONE. After careful consideration, we feel that it has merit but does not fully meet PLOS ONE’s publication criteria as it currently stands. Therefore, we invite you to submit a revised version of the manuscript that addresses the points raised during the review process.

Response 1:

Thank you very much for this positive feedback. We are happy to resubmit a revised version of our manuscript.

Comment 2:

The manuscript has been evaluated by three reviewers, and their comments are available below.

The reviewers have raised a number of concerns that need attention. They request additional information and discussion on how citizen engagement profiles can differ in different types of research to provide context to interpret the results with. They also request additional clarity on the definitions of data collection and data analysis, and note concerns with the small sample sizes for the data analysis profiles and that not all claims may be supported. 

Could you please revise the manuscript to carefully address the concerns raised?

Response 2:

These are valid points from the reviewers. And yes, of course, we have carefully revised the manuscript and have addressed all of the reviewers’ comments accordingly (for our detailed responses, see below).

Reviewer #1:

Comment 3:

Thank you for exploring the extent to which citizens are engaged in different tasks of the scientific process. Many times, citizen involvement in research is based around tokenism, and does not result in meaningful participation of citizens in the work.

Revision suggestions:

Adding some discussion (2-3 lines, maybe a paragraph) around how citizen engagement profiles can differ in different types of research (e.g., health sciences vs. zoological sciences etc) in the background could provide the reader with some context to interpret the results with.

Response 3:

We thank Reviewer 1 very much for the positive feedback regarding our submission.

To provide context for the interpretation of our results, we have now discussed how participants’ engagement potentially differs between projects from specific disciplines in the Introduction section (p. 6).

 “However, even though citizens should be objectively able to participate in the whole scientific process in CS projects [25], it is unclear whether citizens equally engage in all levels of scientific activities or whether they are even interested in doing so [4,9,20]. First, CS projects often take place in informal learning settings [26] which provide citizens with different opportunities to learn from different scientific activities [27]. In this case, their choice of activities may not correspond to the ideal engagement pattern of the highest level [9,19]. Second, the spreading of CS in various disciplines [28] has led to CS projects offering discipline-specific activities for citizens to participate in [21]. For example, in the biological sciences, CS projects often engage participants in observation and monitoring activities [21], while in the health sciences, CS projects may involve their participants in more gamified activities such as digitally folding, and modifying proteins and their structures [29]. Such discipline-specific activities of CS projects might promote that participants’ engagement differs between projects from different disciplines [1].”

Reviewer #2:

Comment 4:

The paper builds on an important issue under examination in the citizen science field-- how to deepen participant-engagement to benefit the citizens and the science. One concern I have is the authors' definition of "data collecting" . Here, it is described as the act of setting out camera traps and identifying/labeling the images captured by the cameras. That actually could be considered data analysis. They are analyzing the data collected by camera traps. The act of making and sharing the observations (by going into nature to take and upload the pictures, for example), is what I expected to see in this paper. Perhaps the authors can address this in the paper a bit? There are thousands of citizen science projects that rely on people to collect and share data and typically that means that the participants make and share observations...not retrieve, analyze and label the data. Again, that seems to be a form of data analysis.

Response 4:

We thank the Reviewer very much for the positive feedback and for bringing this point to our attention. We regret that we have not been clear enough about the reasons why we included species identification in the data collection phase and not in the data analysis phase. To address this point, we have explained our considerations in the Introduction section and the Methods section.

 “The mere collection of data in a CS project is regarded as the lowest level of participation (contributory projects: [19]); collecting data, data analysis, and the interpretation of results in a CS project is seen as an intermediate level (collaborative projects: [19]), because some projects allow for data analyses that go beyond species identification activities [21] and may include statistical data analyses (e.g., [22]); and being involved in the whole research process, from generating research questions to drawing conclusions, is regarded as the highest level (co-created projects: [19]; see also ‘extreme CS’: [20]).” (p. 5)

 “During the data collection phase, participants (1) uploaded camera trap photographs, (2) learned how to identify the wildlife species in the photographs from a tutorial, and (3) identified the species. Participants also (4) identified species in pictures from other participants in order to validate other participants’ identification of species. These tasks of the data collection phase correspond to a task typology that clearly distinguishes observations, species identification, and data entries from data analysis [21] (e.g., statistical data analysis: [22]), although species identification could be seen as some kind of initial data analysis [31].” (p. 9)

Reviewer #3:

Comment 5:

Thank you for conducting this work examining engagement patterns across projects. In general, I think the paper is very strong and is need of minor revisions.

Response 5:

We thank the Reviewer very much for the positive feedback.

Comment 6:

This statement is unclear, please explain: The standard deviation of days between pairs of sequential active days was the variation in periodicity (v).

Response 6:

To be clearer about the calculation of the variation in periodicity, we have rephrased the definition accordingly.

 “We also counted the number of days between each of a participant’s active days and then calculated the standard deviation of these days in between. Thus, we estimated how regularly a participant actively contributed to the project which was the variation in periodicity (v).” (p. 12)

Comment 7:

The sample sizes for the data analysis profiles are so small - can we really say anything about these profiles?

Response 7:

We can understand the Reviewer’s comment. We believe that the strength of our research approach is that we followed all participants of the Wildlife Researcher projects across the two phases of the project, that is, the data collection phase and the data analysis phase. While we agree that the number of the participants in the data analysis profiles is smaller than in the data collection profiles, we also think that it is eminent for our research to show how many participants stayed active after data collection and thus were included in the data analysis profiles. Still, to better highlight the limitations that come with smaller sample sizes, we have added a discussion of the sample sizes for the data analysis profiles to the Limitations section.

 “A second limitation concerns the sample size in our field studies. Especially, the sample sizes for the engagement profiles in the data analysis phase were smaller than the sample sizes of those profiles in the data collection phase. While this difference may limit the conclusion about participants’ engagement in the data analysis, the quantitative and qualitative changes between the profiles also highlight the differences in engagement between the data collection and the data analysis phase. In general, the number of participants was limited to the camera traps available in the CS project.” (p. 32)

Comment 8:

Unsure that this claim is warranted: "We suppose as a post-hoc explanation that ownership (i.e., an individuals’ feeling of possessing a concrete or abstract object or entity; [38]) for the data, that they themselves collected and contributed, might explain why highly active participants also engaged in data analysis." There may be many other factors at play besides ownership. It would be important for you to include alternate explanations (e.g., not interested/motivated to do analysis, feeling "uncredentialed," lack of educational support for analytical skills.

Response 8:

This is a valid point. To make up for it, we have added further considerations on alternate post-hoc explanations such as that those participants who were highly engaged were overall more motivated to participate in both phases. Furthermore, those participants who were highly engaged in the data collection possibly held a specific set of knowledge and skills that also helped them to engage in the data analysis more strongly.

 “We suppose as a post hoc explanation for why highly active participants also engaged in data analysis that those participants felt ownership (i.e., an individuals’ feeling of possessing a concrete or abstract object or entity; [45]) for the data, that they themselves collected and contributed. This could well be the case, as research on project ownership has indicated that high project ownership promotes deeper involvement in problem solving in the inquiry process [46]. Other possible explanations for why highly active participants in the data collection also engaged in data analysis could be that those participants were in general more motivated to participate in all phases [47] or even held a specific set of knowledge and skills that helped them to analyze the data as well [48].” (p. 29)

Thank you very much again for your positive feedback. We hope that we have revised the manuscript in accordance with what you had in mind. The revisions have improved the manuscript and we hope you agree with us that it is ready for publication.

All the best and kind regards,

The authors

---

## [Decision Letter · Decision Letter 1]

26 Sep 2022

I’m fine with collecting data: Engagement profiles differ depending on scientific activities in an online community of a citizen science project

PONE-D-22-09613R1

Dear Dr. Bruckermann,

We’re pleased to inform you that your manuscript has been judged scientifically suitable for publication and will be formally accepted for publication once it meets all outstanding technical requirements.

Kind regards,

Andrea Fronzetti Colladon, Ph.D.

Academic Editor

PLOS ONE

Reviewers' comments:

Reviewer's Responses to Questions

**Comments to the Author**

1. If the authors have adequately addressed your comments raised in a previous round of review and you feel that this manuscript is now acceptable for publication, you may indicate that here to bypass the “Comments to the Author” section, enter your conflict of interest statement in the “Confidential to Editor” section, and submit your "Accept" recommendation.

Reviewer #1: All comments have been addressed

Reviewer #2: All comments have been addressed

Reviewer #3: All comments have been addressed

2. Is the manuscript technically sound, and do the data support the conclusions?

Reviewer #1: Yes

Reviewer #2: Yes

Reviewer #3: Yes

3. Has the statistical analysis been performed appropriately and rigorously? 

Reviewer #1: I Don't Know

Reviewer #2: I Don't Know

Reviewer #3: Yes

4. Have the authors made all data underlying the findings in their manuscript fully available?

Reviewer #1: Yes

Reviewer #2: Yes

Reviewer #3: Yes

5. Is the manuscript presented in an intelligible fashion and written in standard English?

Reviewer #1: Yes

Reviewer #2: Yes

Reviewer #3: Yes

6. Review Comments to the Author

Reviewer #1: Thank you for addressing the comments provided in the first round of revisions. I think this is a strong manuscript and discuss citizen science in an interesting light.

Reviewer #2: (No Response)

Reviewer #3: Thank you for the revision. You have adequately addressed all of my comments and I have no further recommendations.

7. PLOS authors have the option to publish the peer review history of their article (what does this mean?). If published, this will include your full peer review and any attached files.

Reviewer #1: No

Reviewer #2: No

Reviewer #3: No

---

## [Editor Report · Acceptance letter]

28 Sep 2022

PONE-D-22-09613R1 

I’m fine with collecting data: Engagement profiles differ depending on scientific activities in an online community of a citizen science project 

Dear Dr. Bruckermann:

I'm pleased to inform you that your manuscript has been deemed suitable for publication in PLOS ONE. Congratulations! Your manuscript is now with our production department. 

Kind regards, 

on behalf of

Prof. Andrea Fronzetti Colladon 

Academic Editor

PLOS ONE